# Plasma Transthyretin as A Biomarker of Sarcopenia in Elderly Subjects

**DOI:** 10.3390/nu11040895

**Published:** 2019-04-21

**Authors:** Yves Ingenbleek

**Affiliations:** Laboratory of Nutrition, Faculty of Pharmacy, University Louis Pasteur, F-67401 Strasbourg, France; ingen@unistra.fr

**Keywords:** lean body mass, elderly persons, sarcopenia, malnutrition, inflammation, transthyretin, diagnosis, prognosis, outcome

## Abstract

Skeletal muscle (SM) mass, the chief component of the structural compartment belonging to lean body mass (LBM), undergoes sarcopenia with increasing age. Decreased SM in elderly persons is a naturally occurring process that may be accelerated by acute or chronic nutritional deficiencies and/or inflammatory disorders, declining processes associated with harmful complications. A recently published position paper by European experts has provided an overall survey on the definition and diagnosis of sarcopenia in elderly persons. The present review describes the additional contributory role played by the noninvasive transthyretin (TTR) micromethod. The body mass index (BMI) formula is currently used in clinical studies as a criterion of good health to detect, prevent, and follow up on the downward trend of muscle mass. The recent upsurge of sarcopenic obesity with its multiple subclasses has led to a confused stratification of SM and fat stores, prompting workers to eliminate BMI from screening programs. As a result, investigators are now focusing on indices of protein status that participate in SM growth, maturation, and catabolism that might serve to identify sarcopenia trajectories. Plasma TTR is clearly superior to all other hepatic biomarkers, showing the same evolutionary patterns as those displayed in health and disease by both visceral and structural LBM compartments. As a result, this TTR parameter maintains positive correlations with muscle mass downsizing in elderly persons. The liver synthesis of TTR is downregulated in protein-depleted states and suppressed in cytokine-induced inflammatory disorders. TTR integrates the centrally-mediated regulatory mechanisms governing the balance between protein accretion and protein breakdown, emerging as the ultimate indicator of LBM resources. This review proposes the adoption of a gray zone defined by cut-off values ranging from 200 mg/L to 100 mg/L between which TTR plasma values may fluctuate and predict either the best or the worst outcome. The best outcome occurs when appropriate dietary, medicinal and surgical decisions are undertaken, resuming TTR synthesis which manifests rising trends towards pre-stress levels. The worst occurs when all therapeutic means fail to succeed, leading inevitably to complete exhaustion of LBM and SM metabolic resources with an ensuing fatal outcome. Some patients may remain unresponsive in the middle of the gray area, combining steady clinical states with persistent stagnant TTR values. Using the serial measurement of plasma TTR values, these last patients should be treated with the most aggressive and appropriate therapeutic strategies to ensure the best outcome.

## 1. Introduction

The term “sarcopenia” was coined by Rosenberg [1] to define the loss of muscle mass and muscular strength as an effect of aging, underlying malnutrition and/or inflammatory disorders. Sarcopenia is a major symptom of LBM depletion associated with physical disability [2], increased length of hospital stay [3], and mortality risk [4], all likely to have an impact on the outcome of patients. However, many aspects of the physiopathology of sarcopenia have not been fully elucidated and thus require additional tools for the diagnosis and follow-up of muscle composition and function [5]. Moreover, the emergence of sarcopenic obesity and its multiple subclasses has added to the overall confusion, prompting the mandatory need to disentangle the muddle. Given the growing proportion of aging people in westernized societies and the corresponding major public health burden, workers belonging to the National Institutes of Health (Bethesda) have recently deplored the lack of a rapid, simple and inexpensive method to identify sarcopenia in routine clinical practice [6]. The below review is an attempt to address this challenge.

## 2. Body Composition Studies

Several methodological approaches exist in support of body composition studies [7]. A schematic representation of the different tissue compartments is outlined in Figure 1. In a healthy adult man weighing 70 kg, the fat compartment is poorly hydrated and represents 18% of the body weight (BW) [8]. Lean body mass (LBM) contains 20% proteins with a hydration level of 73% in healthy adults of all ages; these proteins are partitioned into an approximately 4 versus 3 proportion between intracellular and extracellular spaces. The mineral mass (MM) compartment represents approximately 11% of the BW and comprises 7% of the elemental abundance with calcium and phosphorus as the predominant components [8]. The use of dual energy X-ray absorptiometry (DXA) for the measurement of ^40^K has emerged as a method of choice for assessing the size of the LBM regardless of age, gender, and disease states. The reliability of this method is grounded on the fact that at least 95% of the naturally occurring nonradioactive potassium (^39^K) remains confined within the intracellular space of all body organs together with minute amounts (0.0117%) of ^40^K, major source of natural β-radioactivity in living tissues [8]. Both ^39^K and ^40^K isotopes show interchangeable turnover in body tissues, maintaining narrow relationships with nitrogenous compounds [9]. The pioneering investigations performed by Forbes [8] constitute the core of our present knowledge on body composition. These data are collected in Figure 2, which represents the evolutionary patterns of total body K (TBK) throughout the whole human lifespan allowing the calculation of LBM in both sexes from birth until very old age [8]. Further studies have described LBM as a composite agglomeration of fat-free tissues that may be metabolically subdivided into a visceral compartment that comprises tissues characterized by rapid turnover rates (liver, intestinal mucosa, thymoleukocytic tissues) and a structural compartment that is distinguished by slower turnover rates (skeletal musculature (SM), skin, connective and cartilaginous tissues, appendages) [10]. The liver, the chief organ of the visceral compartment, weighs approximatively 2.6% of the BW (1.9 kg in healthy adult men) and exhibits an oxygen consumption rate of 44 ml O_2_/kg [11]. The SM, chief organ of the structural compartment represents 38% of BW (approximately 30 kg in healthy adult men) [12] and presents an oxygen consumption rate of 2.3 ml O_2_/kg [11]. The data show that the 20-fold faster protein turnover rate measured in the liver is compensated by the 20-fold heavier SM weight, indicating that the chief organs of both the visceral and structural compartments equally contribute to one quarter (26.4% and 25.6%, respectively) of total body resting energy expenditure (REE), with all other organs contributing to the remaining half. These data were recently confirmed in a clinical investigation revealing that the two predominant organs of both visceral and structural compartments are key determinants of REE, working together to produce 50% of the total body basal metabolic cost [13]. Special attention must be paid to the intestinal mucosa (10% of total REE) [14] and to thymoleukocytic tissues (7% of total REE) [15]. The data indicate that both visceral and structural compartments, working together, generate approximately two-thirds of total body REE. Taken together, these findings support the view that the variability in REE is associated with variation in energy expenditure per kg of tissues of individual organs [16].

The original equipment utilized by the earliest workers were costly and time-consuming, requiring high technical expertise with the risk of radiation exposure. As a result, new methodologies were developed, allowing to distinguish fat mass and lean mass from body mineral content [17] and to assess the regional size of muscle mass [18]. Several other analytical systems have also been designed to quantify the size and content of specific organs or tissues participating in body composition, namely computed tomography (CT), bioelectrical impedance analysis (BIA), ultrasound imaging (USI), and magnetic resonance imaging (MRI). The advantages and disadvantages of each technique for assessing body composition have been extensively detailed in recently published review papers [7,17,18]. Significant progress has been made in the knowledge of sarcopenia in recent years, together with continuing advances in the establishment of standard reference devices and a reduction in experimental time, radiation exposure, and costs.

## 3. Place of Sarcopenia in Body Composition Studies

The causality of sarcopenia is multifactorial, prompting comparative studies with all other LBM components to detect discordant evolving patterns. Kwashiorkor, a (sub)acute deficiency disorder, typically affects young children who flourish during the breastfeeding period but undergo a dramatic deterioration of health at the time of weaning due to the customary consumption of protein-deprived regimens. The morbidity first affects organs belonging to the visceral compartment: hepatic steatosis [19] causing decreased synthesis of liver-secreted protein markers, notably serum-albumin (Alb) [20], transferrin [21], TTR [22], retinol-binding protein (RBP) [23], and insulin-like-growth factor 1 [24]; involutional flattening of the intestinal mucosa [25] which is responsible for malabsorptive syndromes; and a drastic decline in immune capacities [26], rendering these patients highly vulnerable to superimposed infections. In these patients, the muscular mass is relatively well preserved at least in acutely evolving cases. Marasmus is the consequence of protracted dietary deficiencies in most classes of nutrients, proteins and energy affecting all age classes. This nutritional disorder is mainly encountered in population groups undergoing starvation [27] from drought, famine, civil wars, or displacement constraints. In children, the disorder is characterized by severe growth and weight retardation with the disappearance of SM and fat reserves, making these patients emaciated [28]. While there is a decrease in the structural compartment, marasmic subjects experience relatively minor infectious challenges and display well-preserved visceral organs, as documented by moderate plasma drops in hepatic indices [29], sustainable intestinal function, and resilient immune responses. Cancer patients suffering from solid tumors ending in terminal cachexia manifest shrunken SM and tegmental tissues in contrast to the visceral life-supporting system which may be spared to a considerable extent [30]. Acquired immunodeficiency syndrome (AIDS) patients lose one-third of their starting BW during the last months preceding death, and their overall loss of LBM (calculated from the shrinking of their ^39^K body reserves) is proportionately greater than the percentage of BW loss [31].

## 4. Biochemical Assessment of Sarcopenia

In healthy subjects under normal conditions, daily rates of protein turnover (300 to 400 g/day) are energy-requiring processes that maintain balanced homeostasis between protein synthesis and protein breakdown [32]. In adult humans, the daily rate of protein losses reaches 3% of total protein reserves, which are regenerated from different sources: dietary intake (50 to 80 g/day), *de novo* synthesis from endogenous tissue breakdown, and *de novo* conversion from precursor molecules [32]. Along with these adaptive processes, amino acids (AAs) exceeding the actual body needs remain unmetabolized and undergo enhanced ureagenesis (approximately 15–25 g/day) through the urea cycle, whereas minor AA fractions experience oxidation, being released in the urinary output in the form of end-products such as ammonia (approximately 3–4g/day). Minute amounts of free AAs (less than 2 g/day) may also be recovered in the urinary output. Under physiological conditions, urea is thus the main N-catabolite excreted in the urinary output with gender- and age-dependent concentrations reflecting the normal metabolic activities of the visceral compartment. The ureapoiesis may be considerably decreased in most forms of protein nutritional deficiencies [33] or enhanced in most hypercatabolic disorders such as trauma [34], sepsis [35], and thermal injury [36] which are similarly characterized by protein breakdown prevailing over protein synthetic processes [37]. The pathophysiological consequences of these last approaches on muscular size and function are worth investigating because the metabolic role played by SM remains largely underestimated in elderly persons [38].

Creatinine (Cr) is produced by the nonenzymatic dephosphorylative cyclization of phosphocreatine, a normal structural component of the body musculature. The higher androgenic levels of adult men versus adult women explain gender differences in the urinary output of Cr, with higher values found in male subjects. Cr excretion rate is regarded as an expedient indirect method for evaluating body composition in healthy adult subjects [39], as it is positively correlated with the urea appearance rate [40]. In healthy individuals with normal kidney function, the creatinine coefficient is 18 to 21 mg/kg BW in women and 21 to 25 mg/kg BW in men [41], yielding creatininuria values ranging from approximately 1.7 g to 2.4 g/day corresponding to the total body muscle mass. The urinary output of Cr is low at birth, reaching approximately 250–300 mg/day in young children and thereafter increasing linearly until the onset of adolescence, parallel to the growth of SM during the prepubertal period. The concentrations of urinary Cr are severely depressed in those with protein malnutrition but reach the normal values found in healthy children after nutritional rehabilitation [42]. Creatininuria may be regarded as an index of muscle mass that is closely related to REE at all ages [43]. Its stability throughout adulthood demonstrates a 10% decrease for each 10-year increment after the age of 54 years [44] with a steeper slope in elderly men. By the age of 70 years, significant sexual differences are no longer observed for creatininuria. In hypercatabolic states [34,35,36], increased urinary N and Cr output reflects alterations induced by tissue breakdown, which results in a negative AA balance in SM. Falsely high creatininuria levels may be recorded in individuals who consume diets rich in red meat [45].

Urinary 3-methyl-histidine (3-MH) is an isomer of 1-MH known to be a constituent of anserine. In human subjects and mammalian species, SM contains 3-MH in a soluble form as well as a form bound to the muscle proteins actin and myosin [46]. The 3-MH catabolite was measured at levels of 50 mg/day in the urinary output of adult subjects [47] and at approximately 10 mg/day in healthy children [48]. Urinary excretion of 3-MH undergoes a marked decrease in protein-depleted states but is restored to normal values after completion of the refeeding period [48]. Furthermore, 3-MH has been validated as an indicator of SM protein breakdown in most stress disorders such as trauma, sepsis and surgery [49,50]. The 3-MH urinary values may be overestimated after the consumption of meat-rich regimens [51]. The respective usefulness of urinary Cr and 3-MH in clinical studies remains a matter of controversy: whereas some researchers advocate Cr measurement as a more informative indicator of SM than 3-MH [39], other investigators express the opposite opinion [52]. The data indicate that both Cr and 3-MH provide helpful and specific approaches, allowing for the assessment of the evolutionary patterns of SM in most morbid circumstances. Nevertheless, the required surveillance of the dietary context and desirable recovery of 2–3 days of urinary collections preclude widespread application in current medical practice and clinical settings. Cr and 3-MH measurements are more appropriate for research investigations.

## 5. Is the Body Mass Index Correlated to Sarcopenic States?

Garrow and Webster reported, more than three decades ago, the clinical observation that augmentation of the body mass index (BMI) scoring system basically reflects increased accretion of fat reserves in bodily tissues [53]. This concept has been validated using densitometry and anthropometry measurements [54]. Figure 1 shows the changes characterizing the naturally occurring aging process: healthy elderly persons present a physiological tendency to develop their body FM at the expense of the protein and mineral masses [8]. In the 1990s, the World Health Organization (WHO, Geneva, Switzerland) fully endorsed the use of BMI, which permits easy and rapid assessment of both thinness and overweight in clinical and public health practices [55]. The ensuing epidemic of obesity was first observed in the United States following alterations of socioeconomic constraints leading to increased consumption of energy-dense regimens [56]. In the United States, more than one-third of adults older than 60 years have a BMI value above 30, whereas those aged 65–74 years display a BMI of 40.8 [57]. Appearing afterwards in most westernized countries [58], the obesity pandemic is now threatening most affluent cities in developing countries [59]. It was only at the beginning of the new millennium that the WHO recognized the growing epidemic of obesity as a new and major public health problem affecting not only adult and elderly subjects [60] but also children and adolescents [61,62]. The pandemic is transmitted through multinational companies providing highly refined flours devoid of fibers, cheap saturated fats and oils, and soft beverages enriched in sucrose, being also sustained by behavioral changes resulting from reduced physical activity and by frequent snacking. The prevalence of undernutrition remains a serious problem coexisting with the global obesity epidemic in both developing and developed countries [63,64]. As a result, a double burden of diseases has spread worldwide, combining protein shortage with the unavoidable trail of noncommunicable diseases [63,64].

After the seminal discovery of Quetelet’s formula [65] and during the 16 subsequent decades, normal BMI values were regarded as indications of good health and satisfactory nutritional status. Despite the fact that BMI has limited predictive value at the individual level [66], its scoring formula has rapidly expanded after the mounting obesity pandemic worldwide. Large cohorts of clinicians, mainly those working with surgical, cancer, and trauma patients, still rely on BMI for the diagnosis and follow-up of health states [67,68,69]. However, the excessive accumulation of fat reserves in elderly persons, described as sarcopenic obesity, should no longer be regarded as the continuation of a physiological aging process, as outlined in Figure 1, but rather as a distinct, superimposed nosological entity afflicting both sexes and all classes of adults. The disproportionate inflation of fat stores in a rising obesogenic environment had the negative effect to obliterate the nutritional criteria of malnutrition, notably those defining the size and strength of muscle mass. Moreover, the discrepancy between FM and SM data was aggravated by the recently proposed metabolic dissection of adiposity distribution into four different subsets (central, peripheral, android, gynoid) [70], each of which being characterized by specific impacts on risks of complications, length of hospital stay, fatal outcome, and hospital costs. In addition, visceral fat and subcutaneous fat show opposing associations with biomarkers of nutrition and inflammation [71]. Information provided by BMI records is therefore confusing, frequently misleading, inaccurate, and unreliable to assess, prevent, and treat sarcopenia in usual clinical practice. The conclusion generated by this deadlock has prompted a growing number of investigators to dissociate the management of sarcopenia from overweight status [72,73]. Maintaining the binomial association of sarcopenia and obesity is no longer sustainable by current pathophysiological considerations, explaining why Heymsfield himself, together with his Brazilian coworkers, has recently advised clinicians against using the BMI scoring formula [74].

## 6. Measurement of TTR as A Surrogate Biomarker of LBM Components

Given that SM is the chief component of the structural compartment, it seems logical to follow its evolutionary patterns using biomarkers belonging to the LBM compartment rather than screening fat stores. The main obstacle lies in the fact that researchers do not refer to a univocal definition of sarcopenia which appears to be as complicated as the obesity taxonomy. Two recent Korean reviews [75,76] have provided commendable efforts to summarize the multiplicity of proposals expressed in that domain (class I and class II sarcopenia, appendicular lean mass), highlighting the discordances among these indices and the divergent clinical implications generated by each definition in terms of risk factors and health outcomes. A unifying definition for sarcopenia, reconciling divergent and sometimes opposing influences, needs to be reached for the sake of clinical usefulness. Theoretically, all above-described biomarkers (Cr, 3-MH, hepatic indices) might contribute to meet the scoring task, considering that protein reserves constitute the cornerstone of body building. During recent decades, the measurement of plasma TTR has emerged as a promising indicator because this biomarker fulfills all the classical criteria required to define a good biomarker: it is synthesized and produced by the liver, has a small pool size (10 mg/kg BW), circulates confined within the intravascular space, and has a short biological half-life of 2 days [77], thereby readily responding to any fluctuation in protein status. The TTR index allows monitoring all forms of protein-energy malnutrition, from edematous kwashiorkor to emaciated marasmus [78]. In addition, longitudinal studies showed that TTR demonstrates the same age- and gender-evolutionary patterns [79] as those of LBM (Figure 3), indicating that the analyte could operate as a substitute index of LBM [80]. Finally, it is worth recalling that the measurement of plasma TTR is a simple, rapid, and inexpensive micromethod that can be utilized everywhere, even under field conditions or in poor underprivileged countries [78].

Growth, maturation, and decline of muscle mass result from continuing biological processes starting during intrauterine development until the end of life. The growth of the muscle mass is genetically and hormonally programmed, showing in children a linear progression superimposable to that of TTR [81] and devoid of sexual difference until puberty is reached. This last developmental period is followed by a strong S-shaped growth of musculature in adolescent boys which contrasts to a blunted upsurge in adolescent girls [82], this difference being best explained by the respective level of their liver impregnation with androgenic molecules (Figure 3). Adulthood is characterized by maintenance of musculature and TTR values at plateau levels in both sexes until the age of 60–65 years. The adequacy of the current dietary allowances (RDAs) for protein intake in older adults is usually estimated at 0.8 g/kg^−1^d^−1^. This amount seems clearly inadequate in chronic protein-depleted states (inhabitants of developing countries) or in veganism (plant-eating populations) because, on a ponderal basis, vegetable products barely contain half the concentrations of N and methionine (Met) than those measured in animal items [83]. The data imply that plant products do not meet the true requirements of human tissues to allow appropriate protein syntheses. Vegan subjects therefore disclose highly significant (*p* < 0.001) downsizing of their LBM and SM values [84]. Rat experiments have shown that protein restriction downregulates the hepatic production of TTR [85,86]. Similar results are recorded in clinical investigations undertaken in malnourished and/or vegan patients [87]. The data suggest that the hepatic and muscular functional alterations induced by protein restriction may be stratified using declining TTR values to grade the severity of involutional LBM processes.

The dietary context of the roles played by the levels of protein intake and of essential AAs (EAAs) in these maturational events supports major research topics in musculature pathophysiology focusing specifically on 2 EAAs. The first is leucine (Leu) whose molecular mechanisms of action have been progressively clarified. Leu stimulates protein synthesis through the activity of mammalian target of rapamycin (mTOR1) which may be inhibited by a factor exerting general control nonderepressible (GCN2) effects upon Leu deprivation [88,89]. These studies indicate that enrichment of rehabilitation formulas with Leu is a valuable nutritional measure. The second is Met, an S-containing EAA involved in the ribosomal launching of protein syntheses starting with the attachment of a free Met molecule to initiator transfer RNA to yield formyl-methionyl-tRNA which subsequently binds to the 40S ribosomal subunit before initiating mRNA translation [90]. Homocysteine (Hcy) is the precursor substrate of Met in the remethylation (RM) cycle that converts Hcy → Met under the metabolic regulation of folate and cobalamin hydrosoluble vitamins [91]. The former nutrient is amply supplied by green leaves [92] insuring plant-eating populations protection against the risk of developing Hcy-induced folate deficiency. In contrast, the latter nutrient is virtually absent from vegetable products and is responsible for Hcy-dependent cobalamin deprivation [93], which has well-identified cytotoxic effects [94]. Dietary shortage in cobalamin constitutes a current observation in plant-eating countries [95,96] and its therapeutic provision is mandatory to treat nutritional deficits. Nevertheless, some workers have observed that long-term administration of high cobalamin dosages does not necessarily reduce Hcy concentrations to baseline levels [96]. The persistence of elevated Hcy plasma values appears to be controlled independently from cobalamin status and likely results from chronic dietary N- and Met-deficiencies (see below). The significant downsizing of LBM and SM in vegan subjects [84] corroborates the concept of *unachieved LBM replenishment* [97], including among others depletion of N- and S-resources, hence providing strong support on behalf of rising RDA values in protein-malnourished elderly people [98].

## 7. TTR as A Biomarker of Sarcopenia in Elderly Persons

The evolutionary trajectories outlined by the LBM data and those disclosed by TTR plasma values throughout the entire human lifespan are shown here. The findings indicate that the superimposition of K and N parameters measured in Figure 2 reveal strikingly similar distribution shapes as those defining TTR patterns in Figure 3. Starting from this last Figure as a working basis, the below section intends to provide deeper insight into the four last decades of life in healthy elderly people that are characterized by genetically programmed reduction in size and strength of their aging musculature [99,100]. This naturally evolving process is under the control of several hormones [101] maintaining, under steady state conditions, a balance between anabolic and catabolic influences. The involutional process may be temporarily attenuated by diet and physical activity [102] or pharmacological agents [103]. Detailed information is provided on 17,645 elderly persons enrolled in four decennial categories ranging from 60 to 100 years of age (Figure 4), showing that the TTR plasma concentrations of healthy centenarians are 16.3% lower than those characterizing adulthood.

These findings must be compared with those of the other available approaches seeking to assess the degree of sarcopenia. Using CT, BIA, USI, MRI, or DXA methodologies, longitudinal investigations have shown that the musculature of elderly persons does not reveal a noticeable decrease before the age of 60 years [8]. Thereafter, body composition studies demonstrate gradual SM reduction rates amounting to 6% per decade after mid-life [104], reaching a total loss of approximately 20–25% in healthy centenarians. Estimations of the rate of SM loss may nevertheless display large discrepancies ranging from 0.4% to 2.6% per year among cross-sectional studies [105]. The clinical assessment of sarcopenia is currently defined using anthropometric criteria; the most frequently applied formula involves the calculation of the appendicular SM mass (kg/height^2^ (m^2^)) which is less than two standard deviations below the mean of a healthy young reference group. The rate of prevalence has been estimated at 5–10% of persons over 65 years of age [106] and 22.6% and 26.8% in healthy women and men, respectively, aged from 64 to 93 years [107]. The prevalence of sarcopenia may range from 14% to 33% in long-term care population groups [101]. In a study comprising 91 healthy men and 100 healthy women aged 60 and older, Swiss workers observed discordant results between two independent sarcopenic indices, namely appendicular skeletal muscle mass (ASMM) and relative skeletal muscle mass (RSMM), indicating the necessity to reassess the accuracy of the definition of sarcopenia in clinical settings [108]. The data recorded for LBM, TTR, and sarcopenia rates move downward in the same direction and show comparable declining curves. TTR values recorded in our elderly subjects are well above the cut-off value of 200 mg/L regarded as the lower limit of normalcy. The TTR data were collected on volunteers submitted to careful clinical examination and laboratory screening to discard any type of underlying disease. The TTR declining patterns reported in these selected elderly subjects therefore represent a curve of good health that might serve as a reference tool in future investigations. This situation is compatible with prolonged survival and upkeep of basic muscular aptitudes and physical activity when no stress period occurs, reminding us that elderly persons are highly vulnerable due to the lack of protein reserves in their body tissues, incurring therefore the risk of dramatic complications when stress disorders arise. These results suggest that aged persons living under apparently healthy conditions but identified as having plasma TTR values declining below the cut-off line of 200 mg/L should be regarded as suspected of undetected protein malnutrition. This perception would become strongly reinforced in the cases where lowering TTR concentrations are associated with Hcy values rising in the opposite direction (see below).

This period of life is characterized by low grade circulating levels of cytokines oversecreted by activated leukocytes and is usually associated with acute-phase reactants (APRs) [109,110], suggesting that a rampant inflammatory burden may contribute to the silent breakdown of SM of elderly persons. Interleukin-_6_ (IL-_6_) is the major regulator of any declared stress disorder [111] governing most immune and defense reactions, in combination with tumor necrosis factor α, other cytokines, and APRs, mainly C-reactive protein (CRP) [112,113]. The prevailing role played by IL-_6_-induced production is documented by the identification of 36 metabolites positively or negatively associated with log IL-_6_ [114]. An excess of N-catabolites is excreted in the urinary output, causing additional depletion of LBM stores. These findings are consistent with the view that breakdown processes predominate over repair syntheses [37], illustrating the concept of *excessive LBM losses* [97]. It is worth stressing the point that IL-_6_ suppresses the liver production of TTR as well in animal [115] as in clinical [116] experiments. Taken together, these last data suggest that the declining TTR values are matching and occur in parallel with LBM losses recorded during inflammatory disorders, exposing N-depleted elderly persons to stressors once the cut-off line of 200 mg/L is crossed.

In this large reorchestration of metabolic priorities, peculiar attention should be paid to two distinct but mutually interacting aspects of Met metabolism. The first aspect follows the clinical observation that septic [117] and trauma [118] patients excrete large urinary amounts of elemental sulfur (S). The data likely result from cytokine-induced alterations affecting the activity of muscle ribosomal S40 organelles which undergo a 65% inhibition rate under sepsis conditions [119], thereby severely downregulating the production of Met-dependent molecules. Measurement of S and N in the urinary output of trauma patients yields values close to the 1:14 ratio [118] characterizing mammalian tissues [83], indicating that body S stores and LBM follow concomitant breakdown patterns throughout the course of stress. The spillover may affect all S-containing molecules, notably Met-derivatives found in all LBM tissues [120], glutathione mainly generated in the liver [121], cysteine, and taurine prevailing in the liver and muscle tissues [122]. The second aspect results from original investigations performed in intensive care units which revealed that critically ill patients have high plasma Hcy values unrelated to folate and cobalamin deficiencies [123]. The normal Hcy → Met remethylating cycle is regulated by two converting enzymes, namely methionine synthase (MS) and betaine-homocysteine methyltransferase (BHMT) [91]. Animal experiments have shown that BHMT is absent from LBM tissues of mammalian species [124], implying that, under quiescent conditions, MS functions alone to meet appropriate Met requirements in each bodily organ. Under stressful conditions, the RM capacity of MS is clearly overwhelmed, and thus it fails to fulfill the increased Met needs in damaged regions as BHMT manifests focal refractoriness [125,126]. The molecular mechanisms through which the Met cycle is disrupted have been identified in animal experiments using a specific BHMT chemical inhibitor [126], leading to upstream sequestration of Hcy proportional to the duration and severity of the causal factor. Elevated Hcy levels in body fluids operate as precursor pools rapidly taken up by liver parenchymal cells to undergo intrahepatic RM of Hcy to novel Met molecules [127] which are exported back to the site of injury. The RM process is controlled by cystathionine-ß-synthase governing the transsulfuration pathway, which maintains Met homeostasis in LBM tissues [125] at the expense of increased Hcy-dependent toxicity [94]. Inflammatory states causing urinary losses of Met molecules are associated with significant hyperhomocysteinemia [128]. The high Hcy concentrations found in the sizeable percentage of apparently healthy elderly persons [129] are best explained by underlying N- and Met-deficiencies remaining undetected inasmuch as they demonstrate folate- and cobalamin-repletion states. Unsurprisingly, these high Hcy findings are negatively correlated with declining TTR plasma concentrations [87]. It is here suggested that combined measurement of diminished TTR levels and rising Hcy values should constitute a promising tool to appraise the early stages of subclinical protein malnutrition. In a recent review, more than 50% of the population older than 80 years was shown to suffer from a multifactorial geriatric syndrome [130]. The data suggest that substantial reinforcement of protein intake should be given to elderly persons, as strongly recommended by a large panel of nutrition experts [131].

Taken together, these observations suggest that both visceral and structural components may undergo N-depletion and/or N-recovery processes that can impact the hepatic secretory rates of plasma TTR which thus emerges as a very helpful indicator of LBM resources. The stimulatory or inhibitory factors implicated in TTR production likely integrate centrally-mediated regulatory mechanisms governing the balance between protein accretion and protein breakdown as well as interorgan signals between LBM components, as described elsewhere [80]. Available data support the view that the TTR marker might operate as a surrogate indicator of LBM (or of body cell mass, BCM) values in health and disease (Figure 1). One recent study has shown that TTR correlates positively with BCM in predialytic kidney patients [132]. A clinical investigation performed on elderly persons indicated that, among all visceral markers, TTR showed the highest positive correlation (r = 0.64) with LBM (expressed in the form of fat-free mass index) compared with RBP (r = 0.57) and Alb (r = 0.52) [133]. Two independent studies conducted on comparable groups of kidney patients concluded that LBM [134] and TTR [135] are equally informative to grade the morbid process and predict lethal outcome. Based on the classical TTR reference values [79], subjects with TTR concentrations above 200 mg/L are unlikely exposed to harmful complications. Patients with values below the cut-off line of 200 mg/L incur an increasing risk of serious infectious or metabolic events, as predicted in maintenance dialysis [136], retrospective elective spine surgery [137], cardiac surgery [138], and digestive carcinoma [139]. Values below 100 mg/L bear an ominous prognostic significance, as documented in critically ill patients [140], acute kidney injury [141,142], ischemic/hemorrhagic stroke [143], and cancer [144]. These last investigations identify a nadir level likely resulting from the exhaustion of mobilizable LBM resources that participate actively in defense and repair mechanisms and are characterized by rapid turnover rates. Data collected from AIDS patients [31] indicate that these labile reserves represent approximately 40% of LBM proteins, as opposed to the remaining 60% (cytoskeleton, connective, cartilaginous, and MM-supporting structures) that must be regarded as nonexchangeable. The above data appear to provide preliminary results needing to be checked in randomized double-blinded studies to validate the surgical, medicinal and nutritional interventions to be given to those patients whose initial TTR values were between the 200 mg/L and 100 mg/L cut-off values.

## 8. Concluding Remarks

Besides its primary role as carrier-protein of thyroid hormones and RBP, plasma TTR exerts numerous additional functions [145]. The tetrameric TTR molecule secreted by the brain choroid plexus is shown endowed with neuroprotective properties [145]. It is now recognized that normal TTR molecules and more than 100 mutated variants can aggregate in several body tissues (notably heart, liver, and kidneys) undergoing *in situ* amyloidogenic processes leading to involutional morbidities [78,145]. The TTR biomarker is also acknowledged as an early, sensitive and efficient tool to predict the best recovery from any disease process. Dellière and Cynober have recently proposed an algorithmic framework that functions as a practical guide to help the clinicians stratify patients by risk of complications and outcome [146]. In clinical practice, the lowest measurable TTR values may exceptionally fall to 40–50 mg/L, thereby standing beyond any survival expectation. The present review also allows us to bring an old debate to a close. Some workers, while recognizing the unique role played by TTR in the nutritional recovery of protein-depleted states [78] did raise objections against drawing similar conclusions in inflammatory disorders [147,148]. The main argument expressed by these workers is based on the upsurge of cytokines [115,116] causing TTR to drop and to seemingly operate as a negative APR devoid *ipso facto* of any nutritional significance in inflammatory states [147,148]. Recent data, however, have completely nullified that premature opinion through the demonstration that the decline of TTR and RBP in stressful disorders releases into the bloodstream free fractions of thyroxine and retinol functioning as second frontlines strengthening the effects primarily initiated by cytokines [149]. TTR and RBP should no longer be regarded as passive and inert indices of the stress reaction, but rather as contributors actively implicated in inflammatory responses.

## Figures and Tables

**Figure 1 nutrients-11-00895-f001:**
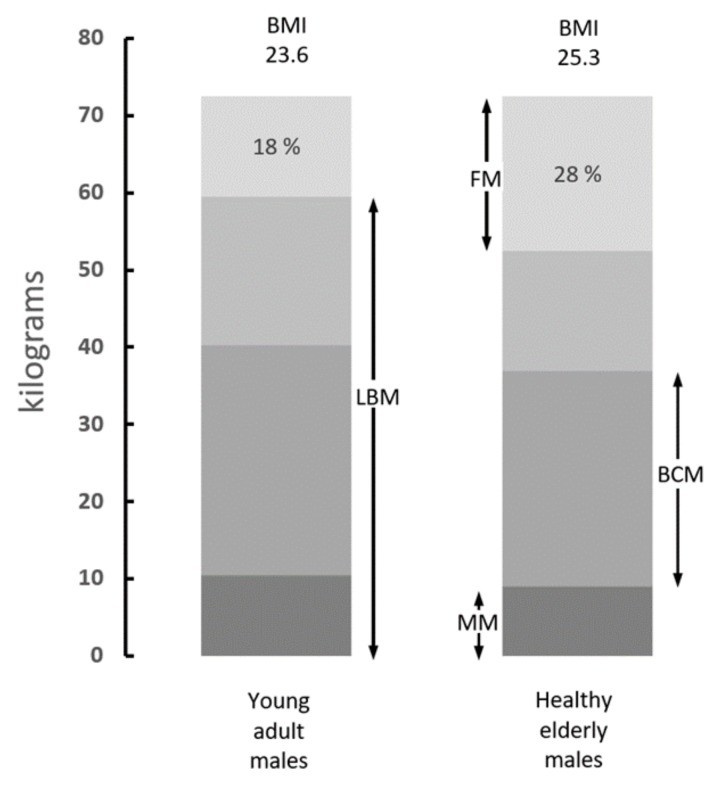
Body composition of average adult and elderly men. Compilation of data from various sources showing the distribution of body tissues in healthy adult subjects (20–25 years) and elderly persons (60–70 years). Fat mass (FM) is poorly hydrated and comprises the bulk of lipid stores. Lean body mass (LBM) is the sum of water, proteins, nitrogenous compounds, and mineral mass (MM). LBM devoid of its MM component yields body cell mass (BCM), which represents 57% of the LBM and comprises tissues with the richest intracellular concentrations of ^39^K and the highest metabolic activities. The data indicate that the normal aging process is characterized by gradual downsizing of LBM, BCM, and MM, whereas FM shows opposite trends rising from 18% to 28% of the body weight (BW) (Adapted from Forbes 1987) [8].

**Figure 2 nutrients-11-00895-f002:**
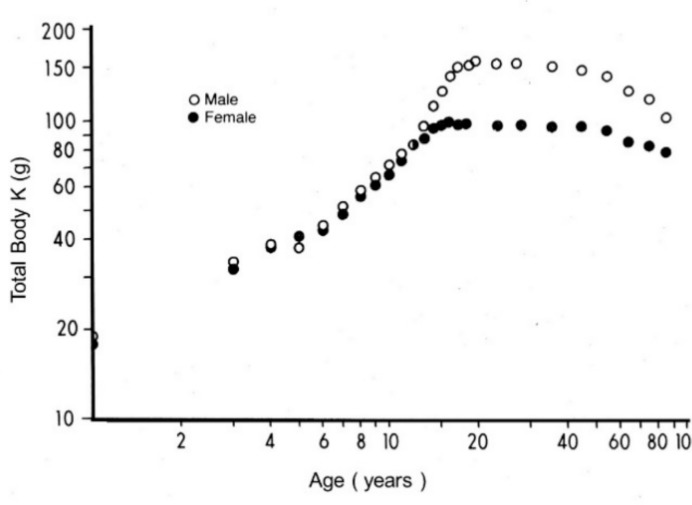
Evolutionary patterns of LBM values throughout the human lifespan. Compilation of seven different clinical investigations performed in healthy subjects from birth to very old age and showing body accretion of TBK levels determined by the measurement of the naturally occurring ^40^K radioisotope using dual-energy X-ray absorptiometry (DXA). The pioneer devices designed to appraise whole body counting and ^40^K in biological tissues were built up at the University of Rochester, Rochester, NY 14642, USA, under the guidance of Gilbert B. Forbes, in close collaboration with the International Atomic Energy Agency, Vienna, Austria. The results are plotted against age on double-logarithmic coordinates. Here, 95% of TBK is sequestered within metabolically active tissues and is narrowly correlated with total body N (TBN), making this last parameter a reliable tool to assess LBM values in health and disease [8]. Starting from the sixties, transthyretin (TTR) reveals a stepwise drop in the process of time with a steeper slope in elderly men that reflects a relatively more rapid decrease in their muscle mass.

**Figure 3 nutrients-11-00895-f003:**
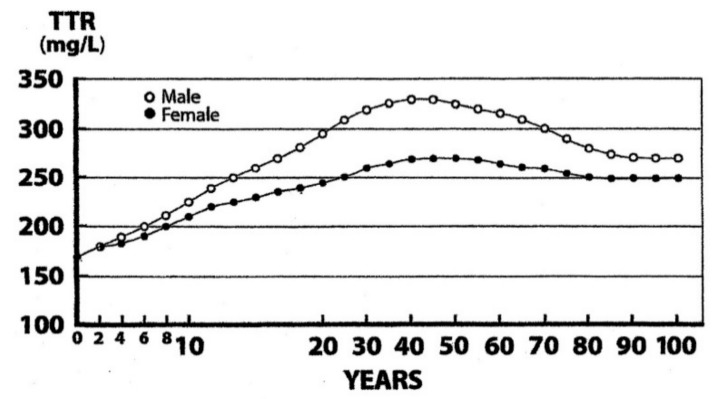
Evolutionary profiles of TTR concentrations throughout the human lifespan. TTR values were measured in the blood samples from 67,720 healthy U.S. citizens using immunoturbidimetric analysis (Bienvenu, 1996) [79]. Equipment and reagents were provided by Beckman-Coulter Inc., Brea, CA 92622, USA and measurements were performed using the skillful expertise of the Foundation for Blood Research, Scarborough, ME 04070, USA. TTR and LBM values manifest closely superimposable trajectories: the lowest values are measured at birth, there is linear increase without sexual difference until the onset of puberty, and there is an occurrence of sexual dimorphism with more pronounced rise in adult males because of a larger muscular size, followed by plateau levels until the age of 65 years, and then a narrowing of the gap between both TTR curves, revealing the disappearance of the sexual difference in the last decades. TTR and LBM curves show comparable abrupt S-shaped elevations at the onset of adolescence until the beginning of adulthood, which are partially obliterated due to alterations in the graduation of abscissa scales.

**Figure 4 nutrients-11-00895-f004:**
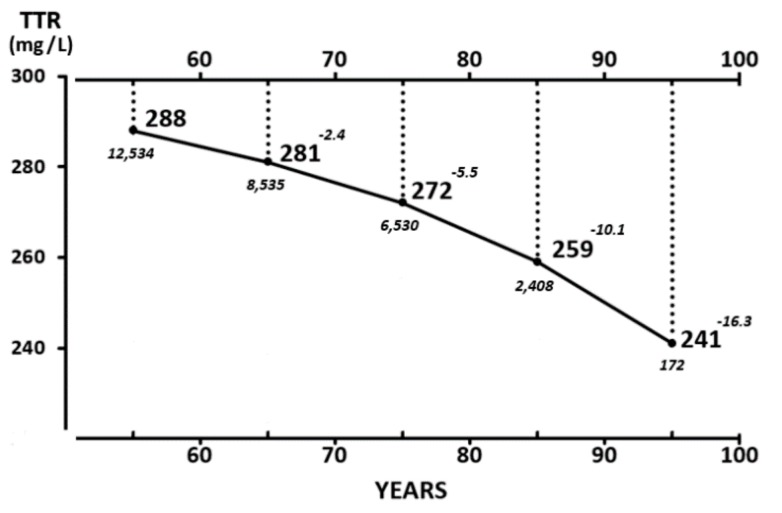
Mean TTR concentrations measured in healthy elderly subjects aged 65 to 100 years. TTR values are measured in 17,645 healthy U.S. elderly citizens (5796 men and 11,849 women) stratified into four groups (61–70, 71–80, 81–90, and 91–100 years of age) [79]. Mean values recorded in both sexes are combined owing to the lack of sexual difference in TTR after 65 years and are expressed in bold characters. The number of participants and increasing deviation from plateau levels in each decennial category are shown in italics. TTR values recorded in both sexes during the last decades are well above 200 mg/L, which is regarded as the lower limit of normalcy, implying that these healthy centenarians, despite their on-going sarcopenia, do not face a life-threatening environment.

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
