# Peer review of "Plasma Transthyretin as A Biomarker of Sarcopenia in Elderly Subjects"

_nutrients, 2019, doi:10.3390/nu11040895_

Round 1
Reviewer 1 Report
The author presents a review regarding transthyretin as biomarker of sarcopenia in the elderly.
Presented data are appropriately presented and of significance for this imminent field of sarcopenia. I think that some minor adjustments need to be conducted.
Abstract:
line 12: clinical studies
line 13: please avoid "wellness"
Introduction:
Could you please add the "revised European consensus on definition and diagnosis of sarcopenia" (Age Ageing 2019) since finally proper functional parameters now can be used for the clinical diagnosis.
Body composition studies:
line 108-112: these lines are not necessary since the histological background is not needed for understanding the TTR parameter.
Place of sarcopenia in body composition studies
line 129: poorly adapted to address nutritional disorders: like whom?
line 139: do you have more clinically relevant examples?
line 141: please avoid idioms.
Biochemical assessment of sarcopenia
line 154: not bold "dietary intakte"
Is the Body Mass Index correlated to sarcopenic states?
line 219-222: Where is this leading? I do not see the importance of this part.
lines 226-227: please avoid conversational language and factoids.
line 229: "double" if you mention four?
line 252: please avoid idioms "final stroke".
Measurement of TTR as a surrogate biomarker of LBM components
line 259/60: please mention the latest definition (see above).
Could you add some molecular insights regarding the role of TTR (Vieira et al. 2014, for example).
287: delete the .
TTR as a biomarker of sarcopenia in elderly persons
line 345: could you give a source for that statement in the plain text, not just the figure?
lines 376-380: these lines are not necessary and far-fetched.
line 431: But these are from 2014, do you have a more recent reference?
line 434: "ultimate indicator" Please avoid idioms.
line 448: "elective spine surgery": please mention that this was just a retrospective consecutive series.
line 456-458: please choose a more conservative formulation, especially since surgical desicion making regarding to plasma ttr is not stated in multiple prospective randomized double-blinded studies.
Concluding remarks
line 468-469: Please shorten this sentence.
Author Response
The author presents a review regarding transthyretin as biomarker of sarcopenia in the elderly.
Presented data are appropriately presented and of significance for this imminent field of sarcopenia. I think that some minor adjustments need to be conducted.
Abstract:
line 12: clinical studies : THE REQUIRED SPACE IS INSERTED ON ( line 12)
line 13: please avoid "wellness" : WELLNES IS REPLACED BY GOOD HEALTH ( lines 16 and 225 )
Introduction:
Could you please add the "revised European consensus on definition and diagnosis of sarcopenia" (Age Ageing 2019) since finally proper functional parameters now can be used for the clinical diagnosis ; THE PAPER BY THE EUROPEAN EXPERT GROUP IS CITED UNDER REF. 5. AND SHORT APPROPRIATE COMMENT IS PROVIDED IN ABSTRACT ( see lines 12-15 ). FURTHER DETAILS ON DEFINITION AND DIAGNOSIS IN THE REVIEW ( lines 347-364 )
Body composition studies:
line 108-112: these lines are not necessary since the histological background is not needed for understanding the TTR parameter : THESE LINES ARE CURTAILED AND CONDENSED ( see LINES, 113-115 )
Place of sarcopenia in body composition studies
line 129: poorly adapted to address nutritional disorders: like whom? : THIS SENTENCE IS DELETED TOGETHER WITH ITS PREVIOUS REFERENCE 18
line 139: do you have more clinically relevant examples? ONE ADDITIONAL REFERENCE PUBLISHED IN NEJM ( 1970 ) BY CAHILL IS GIVEN UNDER THE NEW REF NUMBER 27 ( line 136.)
line 141: please avoid idioms.: IDIOM IS DISCARDED ( line 141 )
Biochemical assessment of sarcopenia
line 154: not bold "dietary intake" : BOLD CHARACTERS UNDERGO LOW-CASE TYPOGRAPHY (LINE 151 ).
Is the Body Mass Index correlated to sarcopenic states?
line 219-222: Where is this leading? I do not see the importance of this part : SENTENCES 219-222 ARE DELETED AND REFERENCE 62 REJOINS REF.61 ( lines 216-219 )
lines 226-227: please avoid conversational language and factoids.: THESE LINES ARE DELETED
line 229: "double" if you mention four ? : THE SENTENCE IS CONDENSED TO DUAL BURDEN ( lines 221-223 )
line 252: please avoid idioms "final stroke". THE SENTENCE ATTRIBUTED TO HEYMSFILD IS CORRECTED ( line 245 )
Measurement of TTR as a surrogate biomarker of LBM components
line 259/60: please mention the latest definition (see above).THE LATEST DEFINITION ARE GIVEN IN REF 5 and. IN THE TEXT
Could you add some molecular insights regarding the role of TTR (Vieira et al. 2014, for example).: THE REVIEW BY VIEIRA PUBLISHED IN BIOMOL CONCEPTS IS BRIEFLY SUMMARIZED IN THE CONCLUDING REMARKS SECTION WITH ADDITIONAL REFERENCE 145 ( see lines 457-462 )
287: delete the TTR as a biomarker of sarcopenia in elderly persons : I AM SORRY, I HAVE NOT UNDERSTOOD THAT QUESTION.
line 345: could you give a source for that statement in the plain text, not just the figure? BESIDES FIGURE 4, LONG COMMENTS ON THE CLINICAL USEFULNESS OF TTR IN SARCOPENIC PATIENTS ARE EXPRESSED ALONG lines 365-377 AND IN A SUMMARY SENTENCE IS GIVEN on lines 423-425 PROPOSING TO COMBINE TTR AND Hcy values.. .
lines 376-380: these lines are not necessary and far-fetched : THESE LINES ARE DELETED..
line 431: But these are from 2014, do you have a more recent reference? IT IS POSSIBLY NOT THE LATEST PAPER ON NUTRITIONAL REQUIREMENTS BUT it STILL KEEPS AUTHORITATIVE AUDIENCE.
line 434: "ultimate indicator" Please avoid idiom :.ULTIMATE IS REPLACED BY VERY HELPFUL ( line 431 )
line 448: "elective spine surgery": please mention that this was just a retrospective consecutive series.
RETROSPECTIVE IS ADDED ( line 445 )
line 456-458: please choose a more conservative formulation, especially since surgical desicion making regarding to plasma ttr is not stated in multiple prospective randomized double-blinded studies. THE ABOVE DATA APPEAR TO PROVIDE PRELIMINAY RESULTS NEEDING TO BE CHECKED IN RANDOMIZED DOUBLE-BLINDED STUDIES TO VALIDATE SURGICAL MEDICINAL AND NUTRITIONAL DECISIONS TO BE GIVEN TO PATIENTS WHOSE INITIAL TTR VALUES WERE BETWEEN THE 200 MG/L AND 100 %MG/L CUT-OFF VALUES ( lines 453-455 )
line 468-469: Please shorten this sentence.: I WOULD LIKE TO MAINTAIN BOTH SENTENCES UNMODIFIED IN THEIR PRESENT FORMULATION SINCE THIS POINT IS OF IMPORTANCE, HAVING BEEN HIGHLY DEBATED ALONG THE LAST DECADE ( lines b470-471 )
Reviewer 2 Report
Overall, this is a clear, concise, and well-written manuscript. Several points make this work interesting. TTR seems a unique biomarker of acutely and chronically disorders. Given the complex phenotypical and pathophysiological frames of sarcopenia and the complexity of aging pose major challenges to the identification of clinically meaningful biomarkers.
Specific comments follow.
Abstract
Brief background and the aim of study were required in abstract.
Introduction
Ι propose an overview of current diagnosis (Diagnostic criteria) of sarcopenia in introduction section
Line 97: Forbes 1987 should appear
Line 25: Again the reference Forbes 1987 should appear and the word- Ref 7 could be deleted.
Biochemical assessment of sarcopenia
Line 154: format- dietary intake
Measurement of TTR as a surrogate biomarker of LBM components
Bienvenu et al., Ref. 79:.keep the number only
Line 273 : Bienvenu et al. should be deleted ,Keep the number only
Line 280: Again Bienvenu et al. should be deleted
Line 347 : Figure instead of FIGURE
Line 349: Adapted from Bienvenu et al., Ref. 79: Again Bienvenu et al. should be deleted
References
Line 500: full name of the author should be added (Filippin)
Author Response
Overall, this is a clear, concise, and well-written manuscript. Several points make this work interesting. TTR seems a unique biomarker of acutely and chronically disorders. Given the complex phenotypical and pathophysiological frames of sarcopenia and the complexity of aging pose major challenges to the identification of clinically meaningful biomarkers.
Specific comments follow.
Abstract
Brief background and the aim of study were required in abstract.: FEEDBACK AND AIM OF THE REVIEW ARE INSERTED IN THE ABTRACT WITH SUPPORT OF REF. 5 IN THE BIBLIOGRAPHY (lines 12-15 )
Introduction
Ι propose an overview of current diagnosis (Diagnostic criteria) of sarcopenia in introduction section : CURRENT DIAGNOSIS IF SARCOPENIA ARE NOT PROVIDED IN THE INTRODUCTION BUT ON LINES 347-364 OF THE REVIEW
Line 97: Forbes 1987 should appear . THE MONOGRAPH BY FORBES IS NOW CITED UNDER REFERENCE 8 IN THE NEW VERSION. FORBES and YEAR OF PUBLICATION ARE CITED IN THE LEGEND OF FIGURE 1 ( line101 ) BUT REFERENCE 8 IS GIVEN ALONE IN THE REST OF THE REVIEW. (see lines 58 to 72 , 109, 207? 350 ). )
Line 25: Again the reference Forbes 1987 should appear and the word- Ref 7 could be deleted.
Biochemical assessment of sarcopenia
Line 154: format- dietary intake : LOW-CASE TYPOGRAPHICAL CHARACTERS ARE INSERTED ON LINE
Measurement of TTR as a surrogate biomarker of LBM components
Bienvenu et al., Ref. 79:.keep the number only SAME AS FOR FORBES. NAME OF BIENVENU AND YEAR ARE GIVEN IN LEGEND OF FIGURE 3 ( line 272 ) AND ALONE ELSEWHERE ( lines 341 )
Line 273 : Bienvenu et al. should be deleted ,Keep the number only
Line 280: Again Bienvenu et al. should be deleted
Line 347 : Figure instead of FIGURE. : CORRECTION MADE WITH LOW-CASE TYPOGRAPHY.( line 3
Line 349: Adapted from Bienvenu et al., Ref. 79: Again Bienvenu et al. should be delete
References
Line 500: full name of the author should be added (Filippin) Filippin’s NAME IS CORRECTED IN REFERENCE 4.
Reviewer 3 Report
The manuscript (number: nutrients-474991), titled: “Plasma transthyretin as a biomarker of sarcopenia in elderly subjects” appears interesting.
Sarcopenia is an age-related, progressive and generalised skeletal muscle disorder that has been associated with increased likelihood of adverse outcomes including falls, fractures, physical disability and mortality. This condition has considerable societal consequences for the development of frailty, disability and health care planning. Furthermore, interest in sarcopenia identification has grown since its recent recognition as muscle disease with an ICD-10-MC Diagnosis Code (Vellas et al., The Journal of frailty & aging. 2018;7(1):2-9). Thus, to identify biomarkers of this clinical condition appear of great interest.
The manuscript is well written. However, I have one general comment.
Speculations about muscle mass are present in the text, but Author did not consider the muscle strength. The consensus document of the EWGSOP on the definition of sarcopenia set the low muscle mass as the essential criterion for its diagnosis together with low muscle strength or low physical performance (Cruz-Jentoft et al., Age and ageing. 2010;39(4):412-23). Recently, EWGSOP revised the document, suggesting that muscle strength is a criterion for the assessment of sarcopenia, and mass quantity or quality are criteria for confirming this condition in clinical practice (Cruz-jentoft et al., Age and ageing. 2019;48(1):16-31). Thus, in my opinion, to speak about sarcopenia only referring to muscle mass without mention muscle strength. Are in the peer-reviewed literature present studies investigating the potential association between transthyretin and muscle strength and/or function?
Author Response
The manuscript (number: nutrients-474991), titled: “Plasma transthyretin as a biomarker of sarcopenia in elderly subjects” appears interesting.
Sarcopenia is an age-related, progressive and generalised skeletal muscle disorder that has been associated with increased likelihood of adverse outcomes including falls, fractures, physical disability and mortality. This condition has considerable societal consequences for the development of frailty, disability and health care planning. Furthermore, interest in sarcopenia identification has grown since its recent recognition as muscle disease with an ICD-10-MC Diagnosis Code (Vellas et al., The Journal of frailty & aging. 2018;7(1):2-9). Thus, to identify biomarkers of this clinical condition appear of great interest.
The manuscript is well written. However, I have one general comment.
Speculations about muscle mass are present in the text, but Author did not consider the muscle strength. The consensus document of the EWGSOP on the definition of sarcopenia set the low muscle mass as the essential criterion for its diagnosis together with low muscle strength or low physical performance (Cruz-Jentoft et al., Age and ageing. 2010;39(4):412-23). Recently, EWGSOP revised the document, suggesting that muscle strength is a criterion for the assessment of sarcopenia, and mass quantity or quality are criteria for confirming this condition in clinical practice (Cruz-jentoft et al., Age and ageing. 2019;48(1):16-31). Thus, in my opinion, to speak about sarcopenia only referring to muscle mass without mention muscle strength. Are in the peer-reviewed literature present studies investigating the potential association between transthyretin and muscle strength and/or function?
THE POSITION PAPER BY EUROPEAN EXPERTS PUBLISHED IN THE JANUARY 2019 ISSUE OF “ AGE & AGEING “ IS CITED IN THE BIBLIOGRAPHY UNDER THE REFERENCE N° 5 . A SHORT COMMENT IS PROVIDED IN THE ABSTRACT OF THE REVIEW ( lines 12-15 ). THE PRESENT REVIEW FOCUSES PRIORLY ON NUTRITIONAL AND METABOLIC ASPECTS OF SARCOPENIA. IT IS EXPLAINED IN THE ABSTRACT ( lines 12-15 ) THAT THE REVIEW PROVIDES ADDITIONAL DATA TO THOSE ADDRESSED BY EUROPEAN EXPERTS IN THE QUOTED POSITION PAPER THAT CONCENTRATES ON MUSCLE MASS AND MUSCLE STRENGTH. BOTH REVIEWS THEREFORE SEEM TO WORK COMPLEMENTARILY.